# Thermodynamic Analysis of Bistability in Rayleigh–Bénard Convection

**DOI:** 10.3390/e22080800

**Published:** 2020-07-22

**Authors:** Takahiko Ban

**Affiliations:** Division of Chemical Engineering, Department of Materials Engineering Science, Graduate School of Engineering Science, Osaka University, Machikaneyamacho 1-3, Toyonaka City, Osaka 560-8531, Japan; ban@cheng.es.osaka-u.ac.jp

**Keywords:** maximum entropy production principle, bistability, Rayleigh–Bénard convection

## Abstract

Bistability is often encountered in association with dissipative systems far from equilibrium, such as biological, physical, and chemical phenomena. There have been various attempts to theoretically analyze the bistabilities of dissipative systems. However, there is no universal theoretical approach to determine the development of a bistable system far from equilibrium. This study shows that thermodynamic analysis based on entropy production can be used to predict the transition point in the bistable region during Rayleigh–Bénard convection using the experimental relationship between the thermodynamic flux and driving force. The bistable region is characterized by two distinct features: the flux of the second state is higher than that of the first state, and the entropy production of the second state is lower than that of the first state. This thermodynamic interpretation provides new insights that can be used to predict bistable behaviors in various dissipative systems.

## 1. Introduction

Many attempts have been made to identify a universal function whose extremum determines the development of a system far from equilibrium. Entropy production characterizes systems during nonequilibrium processes [1,2,3,4,5], and its extrema may be used for determining the system behavior [6,7,8,9,10,11,12,13]. Recently, theoretical and experimental investigations have suggested the maximization of entropy production during nonequilibrium processes (the so-called maximum entropy-production principle, MEPP) [10,11,12,14,15,16]. According to MEPP, when a nonequilibrium system transitions from one state to another, it is characterized by the highest rate of entropy production. Analysis based on MEPP can thus be used to predict the transition point between two nonequilibrium states, such as those observed in the morphological changes of crystal growth, mode changes in droplet oscillation, and pattern changes in thermal convection [12,17,18,19]. For example, in the case of friction phenomena in the flow field, fluid velocity is treated as thermodynamic flux, and the transition point is predicted. For a pendant droplet that changes in the oscillation mode induced by the solutal Marangoni effect with viscous dissipation, the transition point of the oscillation mode is predicted from the intersection of the entropy-production curves determined from the velocity of the oscillating droplet, which is considered as thermodynamic flux [18]. However, dissipative systems far from equilibrium frequently include solutions with several linearly stable branches, i.e., bistable behavior. In such cases, the selected solution depends on the initial conditions, and variational principles based on MEPP would not be required. The prediction of bistable behavior in various dissipative systems is considered an unsolved problem when using variational principles and overshadows the universality of MEPP. This problem is addressed in this study by examining these predictions for a situation involving bistable behavior, i.e., where hexagonal and roll flow patterns coexist during Rayleigh–Bénard convection.

## 2. MEPP

The state of a nonequilibrium system is characterized by the thermodynamic flux J expressed as a function of the driving force of the entire system F, which is proportional to differences in the temperature, concentration, pressure, etc. [12,17,18,19]:(1)$$J_{i}=a_{i}\left(F-b_{i}\right)$$
where $a_{i}$ and $b_{i}$ are constant coefficients.

Assume that there is a transition from state α to β as F increases. 

Each state is described by a local entropy-production σ curve, which is characterized by the product of the local thermodynamic force X and thermodynamic flux J [1]. Thus, the relationship between σ and *F* can be expressed as [12,17,18,19]
(2)$$\sigma_{i}=a_{i}{\left(F-b_{i}\right)}^{2}$$

According to MEPP, a nonequilibrium system develops in a manner that maximizes its entropy production under the given binding conditions [10,14]. The transition point between states α and β corresponds to the intersection of the two σ curves A and not that of the two J lines B. Although the coefficients of F in $\sigma_{i}$ are not strictly equal to $a_{i}$ in Equation (1), there is no change in the intersection. Thus, for simplicity, we use the same coefficients. Before intersection A, the entropy-production curve of state α lies above that of state β, whereas after intersection A, the converse is true. It should be noted that when entropy production is expressed as a function of X, σ(X) cannot distinguish between the nonequilibrium states because all σ(X) are present on a single quadratic curve $\sigma \left(X\right)=LX^{2}$ [19], where L is the phenological coefficient [1]. Furthermore, we cannot predict the transition point nor even understand whether another kind of nonequilibrium state exists in the system when the thermodynamic flux expressed as a function of the driving force can be described only on a single line. Even at equilibrium, if we know the thermodynamic properties of water only in the liquid phase at 1 atm, we cannot predict that water will undergo a phase transition to the gas phase at 100 °C. If we know the dependence of the chemical potential of water in both phases on temperature, we can predict the boiling point of water from the intersection of their two chemical-potential curves.

As shown in Figure 1, we find that intersection A represents the transition point between the nonequilibrium states. However, we wish to understand intersection B of the thermodynamic-flux lines in terms of the physical behavior of the nonequilibrium system. It transpires that this intersection is a starting point for the coexistence of two different nonequilibrium states. When $F>F_{B}$, state β begins to manifest because the thermodynamic flux of state β is higher than that of state α. However, the system mainly comprises state α because σ(F) of state α remains greater than that of state β. Therefore, states α and β coexist until $F<F_{A}$. The thermodynamic flux of state β increases continuously with *F*, so the proportion of state β will increase until it represents a major part of the system. Once $F>F_{A}$, the system consists only of state β.

## 3. Thermodynamic Analysis of Rayleigh–Bénard Convection

To verify this MEPP prediction, precise experimental data are required to calculate the relationship between the thermodynamic flux and driving force of a dissipative system exhibiting bistability. One example is reported in the literature [20], where hexagonal and roll patterns coexisted during Rayleigh–Bénard convection, subject to external temporal modulation of the reduced Rayleigh number $\varepsilon_{0}\equiv \Delta T/\Delta T_{c}-1$, where ΔT is the temperature difference between the bottom and top plates of the water-filled container and $\Delta T_{c}$ is the critical temperature difference for the onset of convection without modulation. Here, ε has the form
(3)$$\varepsilon \left(t\right)=\varepsilon_{0}+\delta \;\cos\left(\omega t\right)$$
where the time *t* and frequency ω are scaled according to the vertical thermal-diffusion time and δ is the amplitude of modulation. The 13-mode Lorenz model proposed by Hohenberg and Swift predicts a positive threshold-shift change in the convection onset from $\varepsilon_{0}=0$ to $\varepsilon_{0}=\varepsilon_{c}$ [21]. Above $\varepsilon_{c}$, the roll patterns that appear through supercritical bifurcation are unstable to the hexagonal patterns (reproduced in the inset of Figure 2a) [21]. This continues as $\varepsilon_{0}$ increases until $\varepsilon_{0}=\varepsilon_{R}$, beyond which the roll patterns are stable. Hexagonal patterns manifest through subcritical bifurcation, first becoming stable at $\varepsilon_{0}=\varepsilon_{A}<\varepsilon_{c}$, which continues until $\varepsilon_{0}=\varepsilon_{B}>\varepsilon_{R}$, where they are unstable to roll patterns. For $\varepsilon_{R}<\varepsilon <\varepsilon_{B}$, both hexagonal and roll patterns are stable. Meyer reported that the bistable region for the range of loop $\varepsilon_{B}-\varepsilon_{R}$ was approximately two orders of magnitude larger than that for the loop between $\varepsilon_{A}$ and $\varepsilon_{C}$ [22]. Therefore, it is easy to resolve the bistable region experimentally.

Heat flux $J_{\mathrm{conv}}$ is a dimensionless quantity given by the ratio of the convective heat flux to the heat flux conducted through the fluid. When ε>0, convective flow occurs and $J_{\mathrm{conv}}$ becomes positive. Meyer, Cannell, and Ahlers performed experimental observations of Rayleigh–Bénard convection and the heat flux, as shown in Figure 2a [20]. They detected the bistable region where both hexagonal and roll patterns were stable. The model quantitatively and qualitatively agrees with the experimental results in the pure-hexagonal $\left(\varepsilon_{c}\leq \varepsilon_{0}<\varepsilon_{R}\right)$, bistable $\left(\varepsilon_{R}\leq \varepsilon_{0}<\varepsilon_{B}\right)$, and pure-roll $\left(\varepsilon_{0}>\varepsilon_{B}\right)$ regions for δ=1.97, where $\varepsilon_{c}=0.132$, $\varepsilon_{R}=0.244$, and $\varepsilon_{B}=0.350$.

Let us begin by predicting the transition point between heat conduction and heat convection using MEPP with static measurements without modulation. This is easy to predict because of the linear relationship between $J_{\mathrm{conv}}$ and $\varepsilon_{0}$ in each state.

Here, $\varepsilon_{0}$ corresponds to the driving force of the entire system F. The values of $J_{\mathrm{conv}}$ may be linearly fitted as a function of $\varepsilon_{0}$ in each state. The heat flux for heat conduction is obviously zero. The best-fit line for $J_{\mathrm{conv}}$ of heat convection with $\varepsilon_{0}>0$ is $J_{\mathrm{conv}}=1.29\left(\varepsilon_{0}-9.42\times 10^{-4}\right)$. The entropy production of the heat-conduction and heat-convection regions is easily calculated from Equation (2), where σ=0 and $\sigma =1.29{\left(\varepsilon_{0}-9.42\times 10^{-4}\right)}^{2}$, respectively. Thus, we obtain the intersection $\varepsilon_{0}=9.42\times 10^{-4}\approx 0$. It is clear that the transition occurs at $\varepsilon_{0}=0$, because of the definition of heat flux wherein the positive value of $J_{\mathrm{conv}}$ signifies the occurrence of heat flux produced by only convective flow; however, it is important that this is predicted using MEPP.

Next, we analyze Rayleigh–Bénard convection subject to external temporal modulation on the MEPP basis. The transition points cannot be distinguished easily because the values of $J_{\mathrm{conv}}$ show rounded bifurcations due to modulation from heat conduction to convection; for the heat-convection region ($\varepsilon_{0}>\varepsilon_{C}$), they align approximately along a single line that changes only slightly in slope. The derivative of $J_{\mathrm{conv}}$ with respect to $\varepsilon_{0}$ enables recognizing the point of change in the slope (Figure 3a).

For $0.13<\varepsilon_{0}<0.21$, $dJ_{\mathrm{conv}}/d\varepsilon_{0}$ increases monotonically with $\varepsilon_{0}$, whereas for $\varepsilon_{0}>0.21$, it remains approximately constant. Thus, we can divide the heat-convection region in two based on the change in $dJ_{\mathrm{conv}}/d\varepsilon_{0}$. The values of $J_{\mathrm{conv}}$ in the two regions can be linearly fitted as functions of $\varepsilon_{0}$, yielding $J_{\mathrm{conv}}=1.108\left(\varepsilon_{0}-0.116\right)$ and $J_{\mathrm{conv}}=1.197\left(\varepsilon_{0}-0.125\right)$ (Figure 3b). The former corresponds to the hexagonal convection and the latter to roll convection. Using the method described above yields two curves for entropy production of the hexagonal and roll flows, respectively: $\sigma =1.108{\left(\varepsilon_{0}-0.116\right)}^{2}$ and $\sigma =1.197{\left(\varepsilon_{0}-0.125\right)}^{2}$. These curves intersect at one point, $\varepsilon_{0}=0.354$, even though the two curves appear to overlap.

This intersection corresponds to a transition point between the hexagons and rolls, and it is in good agreement with $\varepsilon_{B}=0.350,$ where the rolls become stable to the hexagonal patterns in both the experimental and theoretical results. Furthermore, the intersection of the two $J_{\mathrm{conv}}$ lines occurs at $\varepsilon_{0}=0.238$. The value of $\varepsilon_{0}$ is close to a starting point for the bistable region, $\varepsilon_{R}=0.244$, where both hexagonal and roll patterns are stable in the experimental and theoretical results. The actual patterns obtained by Meyer et al. are shown in Figure 4.

For $\varepsilon_{0}>\varepsilon_{R}$, the roll patterns gradually overlap with the stable hexagonal patterns, and coexisting patterns persist until $\varepsilon_{0}=\varepsilon_{B}$. As shown in Figure 3b, the $J_{\mathrm{conv}}$ of the rolls lies above that of the hexagons, whereas σ of the rolls lies below that of the hexagons. For $\varepsilon_{0}>\varepsilon_{B}$, $J_{\mathrm{conv}}$ and σ for the rolls lie above those for the hexagons, and the actual patterns show that only the rolls are stable.

The experimental values of $J_{\mathrm{conv}}$ represent the total heat flux produced by the three directional components of heat flux, i.e., *x*, *y*, and *z*. The vertical component, which is aligned with the temperature difference ΔT between the bottom and top plates, accounts for a large proportion of the total heat flux; hence, it obscures the slight change produced by the horizontal components [19]. Thus, the experimental results do not show the jump in $J_{\mathrm{conv}}$, as indicated by the schematic in Figure 1. If the effect of the vertical component is removed from the total flux and the heat flux is analyzed only in the direction perpendicular to ΔT, the jump in $J_{\mathrm{conv}}$ can be observed, and more accurate transition points may be obtained.

## 4. Discussion

The state with the highest entropy is the state where intensive variables are uniform in the entire system. However, the dissipative system applied by external forces, such as difference in temperature, concentration, and pressure fields, never develops into a uniform state. Under the given binding conditions, in order for the dissipative system to approach more rapidly the state with a uniform field, heat, molecules, and the fluid momentum must transfer faster. In Rayleigh–Bénard convection, the heat transfer of the system changes from heat conduction to heat convection in a hexagonal pattern, and then to heat convection in a roll pattern, and finally to turbulence as the difference in temperature increases. The state of heat transfer changes such that the system more rapidly approaches the uniform temperature field. Therefore, the system should realize the state with higher damping and higher energy consumption by changing the flow state to approach the uniform field more rapidly.

The accuracy of the transition point predicted from our thermodynamic approach depends on the precision of the data on the relationship between the thermodynamic flux and driving force. Thus, the predicted transition point necessarily involves some uncertainty. However, as shown in Figure 2 and Figure 3, in Rayleigh–Bénard convection, the state with the highest entropy production is more stable than a state with lower entropy production. Bistability occurs when a new state has higher thermodynamic flux and the existing state has higher entropy production. It is a significant achievement that Rayleigh–Bénard convection cannot develop against the above rules.

## 5. Conclusions

This study shows that thermodynamic analysis based on entropy production can be used to predict the transition point in the bistable region, i.e., the region where hexagonal and roll flow patterns coexist during Rayleigh–Bénard convection. This addresses a gap in our understanding of MEPP with respect to dissipative systems far from equilibrium. These systems frequently require solutions with several linearly stable branches, and in such cases, the selected solution should depend on the initial conditions. This work may be used to predict bistable dissipative-system behavior in a wide range of applications, such as biological switching [23,24,25], optical switching [26,27], and chemical switching [28,29,30].

## Figures and Tables

**Figure 1 entropy-22-00800-f001:**
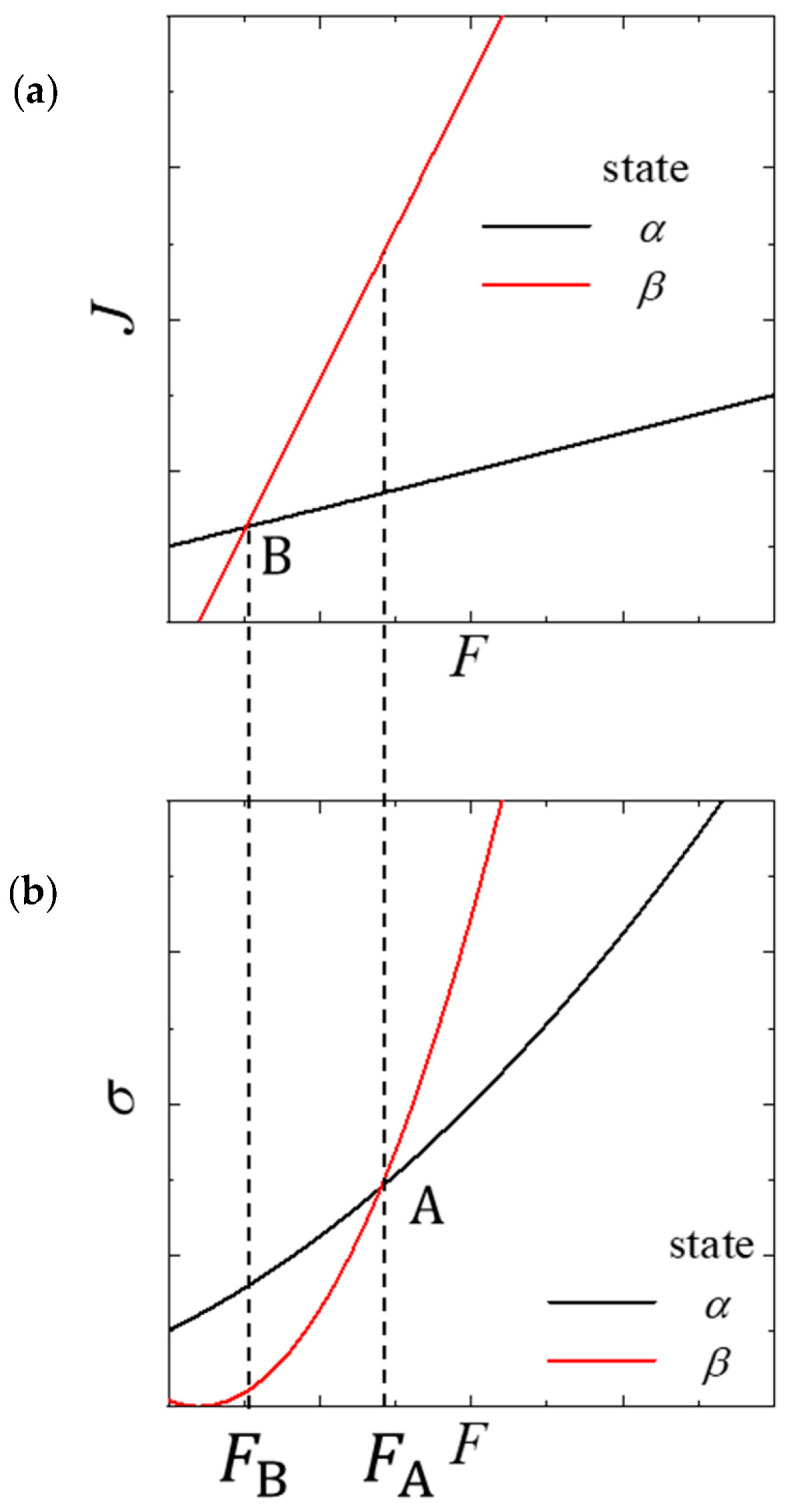
Schematic representation of nonequilibrium transition from state α (black) to β (red). Dependency of thermodynamic flux on the (**a**) driving force and (**b**) entropy production of the driving force. Points A and B are the intersections of the entropy-production curves and thermodynamic-flux lines, respectively.

**Figure 2 entropy-22-00800-f002:**
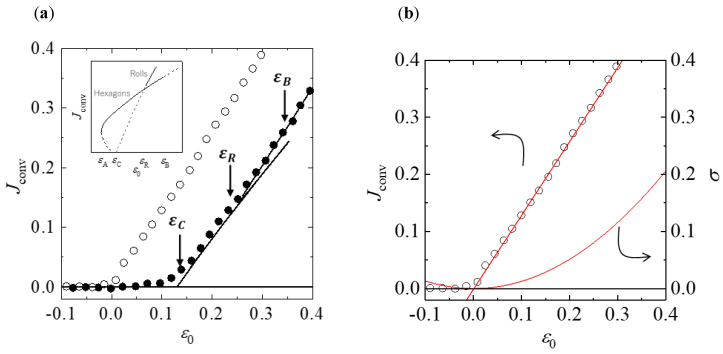
Behavior of the convective heat flux. (**a**) Relationship between convective heat flux averaged over one cycle $J_{\mathrm{conv}}$ and $\varepsilon_{0}$, as presented in [20]. The open circles ○ represent measurements without modulation, and the solid circles ● represent $J_{\mathrm{conv}}$ from measurements taken under modulation, with ω=15 and δ=1.97. The solid curves are predictions of the 13-mode Lorenz model, $\varepsilon_{c}=0.132$, $\varepsilon_{R}=0.244$, and $\varepsilon_{B}=0.350$ [20,21]. Experimental data reported by Meyer et al. [20] for δ=0 and δ=1.97 were used for comparison with the maximum-entropy-production-principle (MEPP) predictions. (Inset) Bifurcation diagram showing the theoretical relationship between $J_{\mathrm{conv}}$ and $\varepsilon_{0}$, as presented in [21]. (**b**) Calculated intersection of the entropy-production curves for the change from heat conduction to convection without modulation. The red line shows the heat flux for convection: $J_{\mathrm{conv}}=1.29\left(\varepsilon_{0}-9.42\times 10^{-4}\right)$. The red curve shows the entropy production for convection: $\sigma =1.29{\left(\varepsilon_{0}-9.42\times 10^{-4}\right)}^{2}$. The black line represents the heat flux and entropy production for conduction: $J_{\mathrm{conv}}=\sigma =0$. The intersection of the two curves is $\varepsilon_{0}=9.42\times 10^{-4}\approx 0.$ Reproduced with permission from [20]; copyright 1988 APS.

**Figure 3 entropy-22-00800-f003:**
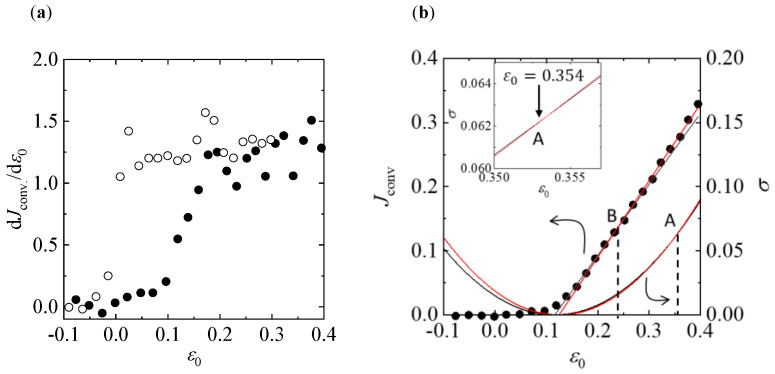
Analysis for bistability. (**a**) Derivative of the convective heat flux with respect to $\varepsilon_{0}$ corresponding to Figure 2a. The open and solid circles represent measurements without and with modulations, respectively. (**b**) Calculated intersection of the entropy-production curves A in the change from hexagons to rolls, and the calculated intersection of the heat flux B where hexagons and rolls coexist, with modulation for ω=15 and δ=1.97. The black and red curves represent the entropy production for the hexagonal and roll flows, respectively: $\sigma =1.108{\left(\varepsilon_{0}-0.116\right)}^{2}$ and $\sigma =1.197{\left(\varepsilon_{0}-0.125\right)}^{2}$. Intersection A is for $\varepsilon_{0}=0.354$. The black and red lines are the heat fluxes of the hexagonal and roll flows, respectively: $J_{\mathrm{conv}}=1.108\left(\varepsilon_{0}-0.116\right)$ and $J_{\mathrm{conv}}=1.197\left(\varepsilon_{0}-0.125\right)$. Intersection B is for $\varepsilon_{0}=0.238$. (Inset) Magnified view of the entropy-production curves.

**Figure 4 entropy-22-00800-f004:**
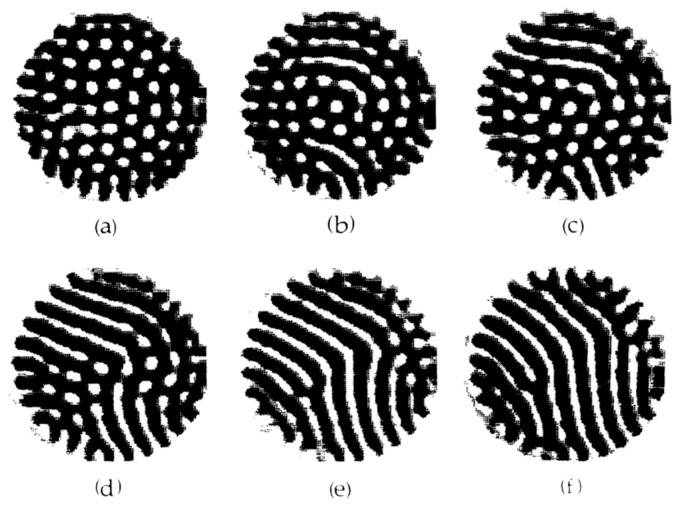
Shadowgraph images of convective flow patterns for ω=15, δ=1.97, and (**a**) $\varepsilon_{0}=0.214$, (**b**) 0.253, (**c**) 0.289, (**d**) 0.325, and (**f**) 0.398. Reproduced with permission from [22]; copyright 1992 APS.

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
