# Peer review of "Thermodynamic Analysis of Bistability in Rayleigh–Bénard Convection"

_entropy, 2020, doi:10.3390/e22080800_

Round 1

Reviewer 1 Report

Dear authors, concerning your paper i have some questions which i think have to be answered in order to make your paper a valuable contribution to Entropy: 1) it is claimed that the state with the highest entroy generation ratio is more stable than a state with lower entropy generation. To determine the intersection of both entropy generation rates, however, they have to be known for both states. How can this be achieved in practice, when only one state (the stable one) can be oberserved? 2) your model for the calculation of the entropy generation rate is very simple. From my understanding it would be inevitable to calculated friction phenomena and losses due to conduction in the flow field (direct approach) or use some kind ov "entropy balance" (indirect approach) at the system boundaries. Can you comment on your very simple approach to compute entropy generation and compare it to the afore mentioned approaches? How are the dissipative losses in the complex flow pattern treated? 3) Figure 3b): The intersection of the two Entropy-generation curved is nor very sharp. Applying some uncertainty around both curves could explain any transition point between 0.1 < e_0 < 0.4 one could say... 4) It is not clear at first glance why a more dissipative process should be more stable, since entropy generation is directly linked to velocity gradients in the system and thus subject to higher damping and a higher energy consumption. Could you suggest a possible conncetion of the first law and the second law of thermodynamics?

Reviewer 2 Report

In this short communication, the author analyses the entropy production of a bistable dissipative system.  The author expresses the thermodynamic flux in terms of a diving force and estimates the entropy production.  Links between the thermodynamic flux and driving forces of a dissipative systems are studied using an experimental example from the literature, Ref [20].  I have to point out that the experimental example is from 1988, and short communications are intended to address more recent developments in the field.  Such advances have been made since 1988 in experiment, theory, and numerical simulations.

Although I am not opposed to the analysis presented in the article, I do not find it relevant anymore.   The author also concludes that the result in the short communication may be used to predict other bistable dissipative systems, including a more recent origami mechanism in Ref [30]; hence, my question is why not directly studying more recent and relevant systems?

I conclude that this study is not appropriate for publishing in its current form as a short communication in Entropy.

Author Response

Responses to Referee 2

In this short communication, the author analyses the entropy production of a bistable dissipative system.  The author expresses the thermodynamic flux in terms of a diving force and estimates the entropy production.  Links between the thermodynamic flux and driving forces of a dissipative systems are studied using an experimental example from the literature, Ref [20].  I have to point out that the experimental example is from 1988, and short communications are intended to address more recent developments in the field.  Such advances have been made since 1988 in experiment, theory, and numerical simulations.

Although I am not opposed to the analysis presented in the article, I do not find it relevant anymore. The author also concludes that the result in the short communication may be used to predict other bistable dissipative systems, including a more recent origami mechanism in Ref [30]; hence, my question is why not directly studying more recent and relevant systems?

I conclude that this study is not appropriate for publishing in its current form as a short communication in Entropy.

Response: Thank you for your comment. Unfortunately, there are no recent experimental studies to provide detailed and quantitative measurements of thermodynamic flux expressed by a function of the driving force in bistable phenomena. We have only limited experimental data in bistable phenomena. Even the more recent origami mechanism in Ref [30] is unavailable to assess my thermodynamic approach. As you pointed out, the experimental example used in this study is not a recent study. However, my manuscript provides new insights into entropy, demonstrates new ideas and proposes a novel use of entropy. I believe that the manuscript is suitable for the scope of Entropy.

Reviewer 3 Report

This paper is well-written and very interesting and gives a clear and predictive explanation for bistability. But I am not totally convinced that the conclusions are correct.

Author Response

Responses to Referee 3

Comments and Suggestions for Authors

This paper is well-written and very interesting and gives a clear and predictive explanation for bistability. But I am not totally convinced that the conclusions are correct.

Response: Thank you for your positive comment. Although you are not totally convinced that the conclusions are correct, but I would sincerely appreciate you making a fair decision based on the findings obtained by this analysis. In Rayleigh–Bénard convection, the state with the highest entropy production is more stable than a state with lower entropy production. Bistability occurs when a new state has higher thermodynamic flux and the existing state has higher entropy production. It is a significant achievement that Rayleigh–Bénard convection cannot develop against the above rules. I hope that the revised manuscript is now suitable for publication in your journal.

Round 2

Reviewer 1 Report

Dear authors,

with the newly intoduced modifications the paper can be published.

Reviewer 2 Report

The author has addressed all my comments.  I recommend publishing of the manuscript in its present form.